# Lower *FGFR2* mRNA Expression and Higher Levels of *FGFR2 IIIc* in HER2-Positive Breast Cancer

**DOI:** 10.3390/biology13110920

**Published:** 2024-11-13

**Authors:** Thérèse Dix-Peek, Caroline Dickens, Juan Valcárcel, Raquel A. B. Duarte

**Affiliations:** 1Department of Internal Medicine, School of Clinical Medicine, Faculty of Health Sciences, University of the Witwatersrand, 07 York Road, Parktown, Johannesburg 2193, South Africa; raquel.duarte@wits.ac.za; 2ICREA and Center for Genomic Regulation (CRG), Dr. Aiguader 88, 08003 Barcelona, Spain; juan.valcarcel@crg.eu

**Keywords:** breast cancer, FGFR2, mRNA expression, alternative splicing, *IIIb*, *IIIc*, HER2

## Abstract

Breast cancer is the most common malignancy in women globally. Genome-wide association studies on breast cancer have found a strong association with fibroblast growth factor receptor 2 (FGFR2). Using publicly available data, we found a higher gene expression of *FGFR2* in the less aggressive luminal cancers, and a lower gene expression in HER2-positive breast cancers. The *FGFR2 IIIb* splice isoform was more common in oestrogen-receptor-positive cancer, while the *FGFR2 IIIc* splice isoform was more common in HER2-positive carcinomas.

## 1. Introduction

Breast cancer, the most common malignancy amongst women worldwide, also- causes the most cancer-related deaths amongst women [1]. It is a heterogeneous disease where the tumours differ greatly in gene expression patterns, growth rates, responses to treatment, and clinical outcomes [2,3]. Since 2000, gene expression profiles have identified five intrinsic molecular subtypes (luminal-A; luminal-B; human epidermal growth factor receptor 2 (HER2)-enriched; basal-like; and normal-like) [4], all of which have different treatment options and clinical outcomes. The luminal-A subtype is generally slow-growing, usually has the best prognosis, and is the most common subtype in European women [2]. Luminal-B subtypes are more aggressive than luminal-A subtypes; the tumours are proliferative and chemotherapy is often offered as a treatment option [2,5]. This subtype is more common in South African, East Asian, and Southeast Asian women [6,7,8,9]. HER2-enriched subtypes are aggressive, but show good results when they are treated with anti-HER2 therapy [2,5]. HER2-enriched subtypes are more common among Asian and African women than European women [6,8,9,10]. The basal-like tumours are usually high-grade with a high proliferation index and have a poorer prognosis and limited treatment options [2,5].

Immunohistochemistry (IHC), using four biomarkers (the oestrogen receptor (ER), progesterone receptor (PR), HER2, and the proliferation marker, Ki67) to distinguish different subtypes, is used as a surrogate for molecular subtyping [3,4,11]. Tumours that are ER+ or PR+, with a low ki67 and null for HER2 are considered similar to Luminal-A, while hormone-receptor-positive tumours with high Ki67 or which express HER2 protein are luminal-B-like. Tumours that are ER−, PR−, and HER2+ are linked to the HER2-enriched molecular subtype, and those that are ER−, PR−, and HER2− are considered to have triple-negative breast cancer (TNBC) which overlaps with basal-like tumours [12,13]. TNBC is the most common subtype in West African women [10].

Mutations in high-risk genes, such as breast cancer association 1 or 2 genes (*BRCA1* or *BRCA2*), have long been linked with breast cancer heritability. More recently, genome-wide association studies (GWASs) have been able to identify moderate-risk alleles in individuals who develop breast cancer and this has led to the detection of over 200 loci associated with breast cancer [14,15,16,17,18,19,20,21]. One of the strongest associations to come out of early GWASs involved variant haplotypes in the fibroblast growth factor receptor 2 (*FGFR2*) [14,15]. *FGFR2* variants have continued to be a top hit in recent GWASs, but are more strongly associated with women of European ancestry and have weaker associations in women from other ethnicities [19,22,23]. The FGFRs are a family of receptor tyrosine kinases that are involved in signalling pathways with roles in a variety of biological processes, including development, cell growth, survival, differentiation, angiogenesis, tumorigenesis [24,25,26], and epithelial–mesenchymal transition (EMT) [27].

During EMT, epithelial cells lose their characteristic features, such as cell–cell junctions, the cobblestone morphology, and the apical–basal polarity, and become more spindle-like with an increased capacity for migration. The cells can degrade the extracellular matrix, invade surrounding tissues, and display increased resistance to apoptosis [28]. EMT is regulated at many levels, including alternative splicing, with the epithelial splicing regulatory proteins (ESRPs) and RNA-binding Fox-1 homolog-1 (RBFOX1) playing major roles in implementing this program [29,30]. The alternative splicing of two exons of the *FGFR2* gene, namely, exon 8 and exon 9, generate epithelial-specific and mesenchymal-specific isoforms, respectively (Figure 1) [30,31,32].

Structurally, the FGFR protein consists of an extracellular domain with three immunoglobulin (Ig)-like loops (Ig I–III), a transmembrane domain, and two intracellular tyrosine kinase (TK) domains (Figure 1) [31,32,33,34]. The Ig loop III is at the core of the FGFR ligand-binding sites, and, as such, rearrangements of this domain affect the ligands that can bind to, and activate, the receptor [35]. In *FGFR2*, the alternative splicing in the Ig III region results in the first half of the loop being formed by exon 7, and the second half of the loop being formed by either exon 8 (*FGFR2 IIIb* isoform, expressed in epithelial cells) or exon 9 (*FGFR2 IIIc* isoform, expressed in mesenchymal cells) (Figure 1). In cancer, the switch from *IIIb* to *IIIc* is one of the criteria of EMT and can cause a metastatic phenotype in carcinomas [28,35,36,37,38].

In breast cancer, FGFR2 promotes hormone-independent tumour growth and progression, the inhibition of apoptosis, and resistance to endocrine therapies [39,40,41]. The RAS/MAPK and PI3K pathways were activated in FGFR2-amplified cancers [42,43]. FGFR2 is usually implicated in ER-negative breast cancer [44], although a low FGFR2 expression is associated with higher proliferation markers and a higher tumour grade [45]. Patients with higher FGFR2 levels had higher survival rates, and were more likely to have ER+ carcinomas [45].

The *IIIb* and *IIIc* isoforms of FGFR2 allow for the specificity of the ligands that can attach to FGFR2, and affect the activity of the intracellular tyrosine kinase. FGFR2 *IIIb* may promote differentiation and apoptosis [26]. In cancer, alternative splicing is often dysregulated and there may be complex switching between the isoforms. FGFR2 *IIIc* overexpression correlates with tumour growth and metastasis [26,46]. GWASs have identified many single-nucleotide polymorphisms associated with breast cancer in intron 2 of *FGFR2* [14,15,17,47], and it has been hypothesised that the acetylation of the histones may affect access to the sites, possibly regulating downstream splicing sites, and generate the *FGFR2 IIIc* isoforms [40,48].

Given the various complex roles that FGFR2 plays in EMT, and its high association with breast cancer in GWASs, we wish to investigate its expression in breast cancer. Using data from The Cancer Genome Atlas (TCGA) (https://www.cancer.gov/tcga, accessed on 15 August 2024), we investigated the associations of breast cancer molecular subtypes and IHC markers with *FGFR2* mRNA expression and *FGFR2* splice isoforms.

## 2. Materials and Methods

### 2.1. FGFR2 Gene Expression Levels in Breast Cancer

Using cBioPortal for cancer genomics (https://www.cbioportal.org/ accessed on 15 August 2024), breast cancer studies that included messenger RNA (mRNA) expression data were interrogated for *FGFR2* expression. Breast Cancer (SMC, 2018; *n* = 187) [49], Breast Invasive Carcinoma (TCGA, Firehose Legacy; *n* = 1108), and Breast Invasive Carcinoma (TCGA, Nature 2012; *n* = 825) datasets were utilised [50].

Plot data of log-transformed *FGFR2* gene expression levels were downloaded in .csv format for prediction analysis of microarray 50 gene assay (PAM50) subtypes (luminal-A, luminal-B, HER2-enriched, and basal-like); immunohistochemistry subtypes (ER+, ER+ HER2+, HER2+, and TNBC); HER2 immunohistochemistry score, ER status, and histological subtype (invasive ductal and invasive lobular carcinomas). *FGFR2* mRNA expression data were downloadable for 519 samples with a PAM50 subtype (from TCGA, Nature 2012; 8 samples with normal-like subtype excluded), 163 samples with an immunohistochemistry subtype (from SMC, 2018; 5 samples with unknown subtype excluded), 622 samples with a HER2 score (from TCGA Firehose Legacy), 1052 samples with ER status (from TCGA Firehose legacy; 2 samples with indeterminate status excluded), and 1046 samples with invasive ductal carcinoma (IDC) or invasive lobular carcinoma (ILC) histological subtype (from TCGA Firehose Legacy).

Data were imported into STATA v14.2 (StataCorp, College Station, TX, USA). A Shapiro–Wilk normality test was performed and data subsequently analysed non-parametrically. *FGFR2* expression levels were represented as box and whisker plots with outliers excluded. Boxes showed medians, and upper and lower quartiles, while whiskers showed non-outlying maximum and minimum values. A Kruskal–Wallis equality-of-populations rank test was used to compare expression levels followed by Dunn’s test of multiple comparisons to compare expression levels between each group. Multivariable linear regression analysis was used to explore confounder-associated *FGFR2* mRNA expression levels in the various breast cancer subtypes. A Bonferroni correction for multiple-comparison testing was applied and *p*-values of less than 0.01 were considered significant.

### 2.2. FGFR2 Transcript Variants

Splice variants of the *Homo Sapiens FGFR2* gene were downloaded from Ensembl Genome Browser 108 (https://www.ensembl.org/index.html accessed on 20 August 2024) along with the reference genome sequence (GRCh38: CM000672.2). *FGFR2* has 41 transcript variants listed in Ensembl, of which 25 are protein coding; 1 is a non-stop decay transcript; 2 undergo nonsense-mediated decay; 5 have non-defined protein coding sequences (CDS); and 8 have retained introns (Appendix A). Transcript variants were aligned to the genome reference consortium human build (GRCh38) using the multiple sequence alignment tool, MUSCLE (https://www.ebi.ac.uk/Tools/msa/muscle/ accessed on 20 August 2024). Aligned sequences were visualised using GeneDoc v2.7, a multiple sequence alignment editor (https://github.com/karlnicholas/GeneDoc accessed on 20 August 2024) [51].

### 2.3. FGFR2 Splice Variants

The TCGA Splicing Variants Database (http://tsvdb.com/plot.html accessed on 6 December 2022) [52] was used to download data on the expression levels of the *FGFR2* gene and *FGFR2* exons in breast cancer patients along with their clinical oestrogen, progesterone, and HER2 receptor status. Samples included were restricted to those of female gender. Size and location of each exon were compared to that of the transcript variants downloaded from ensemble.org to correlate the TSVdb exon numbers to those reported in the literature. Data were imported into STATA v14.2. For each sample, expression levels of individual exons were normalised to the overall expression of the *FGFR2* gene. A Shapiro–Wilk normality test found these normalised expression levels of the exons to be non-normally distributed and, thus, results were presented non-parametrically as box and whisker plots. Boxes showed medians, and upper and lower quartiles, while whiskers showed non-outlying maximum and minimum values. Expression levels between the two exons of interest were compared using a Wilcoxon rank-sum test. A Bonferroni correction for multiple-comparison testing was applied and *p*-values of less than 0.005 were taken as significant.

Splicing patterns of the *FGFR2* gene in breast cancer were interrogated using the TCGA SpliceSeq database (https://bioinformatics.mdanderson.org/TCGASpliceSeq/singlegene.jsp accessed on 13 October 2022). Percent-spliced-in (PSI) values were downloaded for all types of splicing events with filters removed. Information from the TCGA SpliceSeq “Exon Table” was used to compare the sequence and location of the TCGA SpliceSeq exons with the transcript variants downloaded from GRCh38 and, thus, correlate the TCGA SpliceSeq exon numbers with those reported in the literature. TCGA SpliceSeq exon 14 correlated to GRCh38 exon 7, TCGA SpliceSeq exon 15 to GRCh38 exon 8, TCGA SpliceSeq exon 16 to GRCh38 exon 9, and TCGA SpliceSeq exons 17.1 and 17.2 correlated to GRCh38 exon 10 (Figure 2).

Using the TCGA SpliceSeq database, PSI data were transposed and imported into STATA v14.2. Three exon-skipping splicing events were identified as being of interest (Table 1). Analysis was restricted to samples from tumour tissue only. A Shapiro–Wilk normality test showed data for these splicing events to be non-normally distributed and, thus, results were presented non-parametrically as medians and interquartile ranges.

## 3. Results

### 3.1. FGFR2 mRNA Expression and Subtypes

Comparing the *FGFR2* gene expression in the breast cancer subtypes, the mRNA expression was highest in luminal-A and luminal-B tumours, slightly lower in basal-like tumours, and significantly lower in HER2-enriched tumours (Figure 3A). The effect of the patients’ clinical attributes was assessed using a linear regression model. After adjusting for the age at diagnosis, nodal status, tumour stage at diagnosis, and survival outcome, the HER2-enriched molecular subtype was the only factor significantly associated with *FGFR2* expression levels (Appendix A). Similarly, using IHC, the *FGFR2* expression levels in HER2-positive tumours were significantly lower than in ER+ or TNBC tumours (Figure 3B). The regression analysis of IHC subtypes with *FGFR2* showed a significant association with HER2+ tumours, while age and clinical stage were not associated with *FGFR2* (Appendix A).

*FGFR2* expression was higher in ER+ breast carcinomas than ER− cancers (Figure 4A), confirming what was seen in the subtype analysis. Similarly, when tumours were HER2-negative (HER2: 0 or 1+), or equivocal (HER2: 2+), the *FGFR2* expression levels were similar, but the levels were significantly lower in HER2-positive (2+ with FISH or 3+) tumours (Figure 4B). There was also an interesting finding with the tumour type, with the more aggressive IDC having a lower *FGFR2* expression than ILC; patients who had both ILC/IDC components to their carcinomas had *FGFR2* levels between IDC alone and ILC alone (Figure 5). The median and IQR values of the *FGFR2* expression extracted from cBioPortal data plots are shown in Appendix A.

### 3.2. Expression of FGFR2 Exons (Exon 8 and Exon 9) with IHC

Using the TSVdb database, the expression levels of *FGFR2* exon 8 (indicative of *FGFR2 IIIb* expression) were higher in ER+ than ER− tumours (Figure 6). The more aggressive HER2+ tumours had higher levels of exon 9 (indicative of the *FGFR2 IIIc* isoform) (Figure 7), while TNBC had lower *IIIb* and *IIIc* expression levels compared to ER/PR-positive cancers (Figure 8). The relative *FGFR2* exon 8 and exon 9 expression data (proxies for the *IIIb* and *IIIc* isoforms) are shown in Appendix A.

### 3.3. Percent Splicing of Exon 8 and 9 (FGFR2 IIIb and IIIc Isoforms)

The TCGA SpliceSeq data were used to investigate splicing events within the *FGFR2* gene focusing on exons 8 and 9. Percent-spliced-in (PSI) values quantify spicing events, and, in the case of an exon skipping event, they measure the fraction of reads that include a particular exon divided by the total number of reads spanning that region. Since the remaining reads do not contain that exon, these represent the number of reads in which the exon is excluded or ‘skipped’. The *FGFR2 IIIb* and *IIIc* isoforms skip exons 9 and 8, respectively; thus, we looked at the proportion of reads excluding these exons (percentage-spliced-out (PSO)) as our model. In tumour tissue, the proportion of reads that excluded exon 9 (isoform *IIIb*) was 99.1% while the proportion of reads that excluded exon 8 (isoform *IIIc*) was 44.8% (Figure 9). This indicates that there is a substantially higher number of reads representative of the *IIIb* variant than of the *IIIc* variant. In addition, the exon 9 skipping event (Figure 9A) was covered in a substantially greater number of samples (*n* = 1098) than the exon 8 skipping event (Figure 9B), which was only covered in 103 samples.

## 4. Discussion

Aberrant FGFR expression has been reported in many cancers, including cholangiocarcinoma [53], sarcomas [24], keratinocytes [54], and endometrial [55], prostate [30], lung [37], colorectal [56], and breast [57] carcinomas. Classic FGF/FGFR pathways include Ras/Raf-MEK-MAPK (mitogen-activated protein kinases), phosphatidylinositol-3-kinase/protein kinase B (PI3K/AKT), phospholipase Cƴ (PLCƴ), and signal transducer and activator of transcription (STAT). These pathways are affected by ligand-receptor specificity, expression, and alternative splicing [34]. Our study investigated in silico the associations between *FGFR2* expression in breast cancer subtypes using publicly available TCGA data, and further interrogated the *FGFR2* exon 8 and exon 9 expression levels, as proxies of the *FGFR2 IIIb* and *IIIc* isoforms, in clinical subtypes.

Our results pointed to the higher *FGFR2* expression in luminal tumours and basal-like tumours, and the lowest expression levels in HER2-enriched tumours. Similarly, the highest levels of *FGFR2* were in ER+/HER2− subtypes, slightly lower in TNBC subtypes and the lowest in HER2 (ER+ or ER−) IHC subtypes. Bryan et al. found *FGFR2* was underexpressed in breast tumours compared with normal breast tissue [58], while Cox et al. found an association between ER and FGFR2 in HER2-negative tumours, but the association was far weaker in HER2-positive tumours [59]. A further interrogation of our data showed that the expression of *FGFR2* mRNA was higher in ER+ than ER− breast cancers. This agrees with other published data where the FGFR2 expression was higher in breast tumours that are ER+PR+ and ER+PR− compared to ER−PR− tumours [41,45], although FGFR2 has been found to be high in TNBC as well [39], which can also be seen in our study. Overall, *FGFR2* mRNA levels are lower in HER2+ tumours and higher in hormone-responsive tumours.

The interaction between hormones and FGFR2 is established during normal mammary development. Both oestrogen and progesterone are involved in the development of the mammary gland, where oestrogen is involved in ductal formation and progesterone promotes lobular growth [60]. Progesterone and PR work with oestrogen and ER for the expansion of glandular structures during breast development [41]. The FGFR2/FGF10 complex controls early stages of the hormone-dependent development of the mammary ducts, and the survival and proliferation of postnatal luminal and basal epithelial cells [41]. A low FGFR2 is associated with a higher grade, higher Ki67 proliferation index, and worse overall survival [45]. The activation of the FGFR pathway leads to the resistance of ER-directed therapy against ER+ breast cancers, that can be overcome by FGFR pathway inhibitors [41,61,62,63]. The change in PR activity by FGF7/FGFR/JunB also changes the response of ER+ breast cancer cells to anti-ER therapies [64].

A low FGFR2 may be better for tumour progression because ligands that would otherwise bind to FGFR2 in the ECM now bind to other receptors, initiating different pathways. For example, FGF10 causes changes in FRS2/ERK1/2 phosphorylation through the stimulation of FGFR2 [65]. In the absence of FGFR2, FGF10 binds FGFR1, causing an increase in the FGFR1 pathway [43]. Czaplinska et al. have found that FGFR2 levels are higher in normal breast tissue compared to breast carcinomas, and our results suggest that FGFR2 levels are higher in luminal subtypes than the more aggressive TNBC, and the lowest in HER2-positive cancers [66].

In addition to hormone receptors, HER2 is also associated with FGFR2 in breast cancer. We found HER2 expression was associated with a lower *FGFR2* mRNA expression, whether measured by IHC or intrinsic subtyping. Braun et al. [45] found that a low FGFR2 was associated with poorer overall and disease-free survival compared with high FGFR2 levels. The activation of FGFR2 decreased the efficacy of HER2 therapy [67], possibly caused by a switch from the ER/HER2 signalling pathway to the FGFR2 signalling pathway [68]. FGFR2 levels have been found to be amplified and overexpressed in lapatinib-resistant HER2-positive breast cancer cells [69,70]. HER2+ breast cancer cells that are lapatinib-resistant have been found to overexpress FGFR2 levels [69]. Moreover, mouse mammary tumour models have shown that combination therapies against Fgf and *ErbB2* (HER2) lead to increased anticancer effects [57,71].

FGFR2 has been reported in the progression of ER-negative luminal ductal carcinomas, which are more aggressive and less responsive to treatment, suggesting that FGFR2 may have an effect promoting IDC progression to an ER-independent basal-like phenotype [41,61]. TNBC cells with an FGFR2 overexpression have a high susceptibility to apoptosis by the FGFR inhibitor, PD173074 [39]. Zhao et al. [34,72] found the FGFR inhibitor, AZD4547, impaired ductal branching and stem-like characteristics in mammary epithelial and tumour cells in mice. The monoclonal antibody, Bemarituzumab (FPA144), is specific against the FGFR2 *IIIb* isoform, and has been found to inhibit cancerous growth in rats, monkeys, and humans, particularly in gastroesophageal adenocarcinoma [34,73].

The *FGFR2* expression was higher in ILC, and lower in the more aggressive IDC. The activation of the FGF7/FGFR2 axis may cause the degradation of the ER and thus prevent tamoxifen from inhibiting ER+ breast cancer [74]. The FGF10/FGFR2 axis was reported to counter oestrogen-triggered ER+ signalling, thus reducing the effects of anti-ER therapies [41,62]. The FGF/FGFR axis was also involved in PR degradation in breast cancer cells [41]. Patients with IDC have an inverse relationship between the FGFR2 and ER [70,74]. The FGFR2 *IIIb* and *IIIc* splice isoforms differ in their ligand-binding preferences [75]. For example, FGF7, FGF10, and FGF22 bind exclusively to FGFR2 *IIIb* [31,75,76], with FGFR2 *IIIb*/FGF10 being highly associated with breast cancer development [41,77]. Splice switching from FGFR2 *IIIb* to FGFR2 *IIIc* is implicated in tumour progression [26].

Alternative splicing is a post-transcriptional process that leads to alternative mRNA transcripts that encode structurally different protein isoforms [78]. FGFR2 has two mutually exclusive isoforms, FGFR2 *IIIb* and FGFR2 *IIIc*, which have different structures, bind different ligands, and are expressed exclusively in breast epithelial and mesenchymal cells, respectively [78,79]. Functional *FGFR2* mRNAs include either exon 8 to produce the *IIIb* isoform or exon 9 to produce the *IIIc* isoform [33,80], in a tissue-specific manner. A skipped product lacking both of these exons and a double-inclusion product that has both these exons have also been described. These mRNAs encode non-functional receptors that, due to the generation of premature stop codons, are degraded by the nonsense-mediated decay (NMD) mRNA degradation pathway [81]. Tumours are heterogenous, and it is possible to have cells that express the *IIIb* isoform and separate cells that express the *IIIc* isoform in the same tumour. It is also possible that carcinomas cause such perturbations of normal cell behaviour and expression that both the *IIIb* and *IIIc* isoforms may exist in the same cell. A limitation of this study is that, with the data we had available, we were unable to directly compare 7-8-10 (*IIIb*) with 7-9-10 (*IIIc*). However, using PSO levels, we found higher levels of 7-8-10 (suggestive of *IIIb* isoform) than 7-9-10 (*IIIc* isoform).

ER-positive tumours had a higher exon 8 (*IIIb*) expression than ER-negative tumours. FGFR2 *IIIb* is associated with epithelial cells, which have better survival rates. More research had been carried out in the treatment of FGFR2 *IIIb* cancers [41,61,62,63,64]. FGFR2 *IIIc* has not been extensively reported in normal human tissue, but it has been associated with various cancers, including prostate, ovarian, gastric, and breast cancer [46]. The loss of FGFR2 *IIIb* expression is associated with the activation of FGFR2 *IIIc* expression and a shift toward more aggressive behaviour [46]. Our results do not show an increase in exon 8 (*IIIb*) in HER2-positive compared with HER2-negative carcinomas. However, there is a significant increase in the expression of exon 9 (*IIIc*), suggesting an increase in mesenchymal cells in HER2-positive cancers.

The TNBC had *FGFR2* mRNA similar to the ER+HER2− tumours. This is consistent with the results from Lei et al. [82] working in a mouse model that found FGF/FGFR2 signalling drives the development of TNBC as well as the epithelial to mesenchymal transition through FGFR2-STAT3 signalling. Moreover, they found that an increased Fgfr2 resulted in a lower Brca1, promoting tumorigenesis in basal-like mammary tumours [82]. In humans, FGFR-mediated interactions between luminal IDC and its microenvironment can lead to the progression of hormone receptor independence [41].

Therapies against FGFR2 are being investigated in various cancers. However, to date, no FGFR inhibitors have been cleared for clinical use [79,83]. Indeed, FGFR2 therapy in ER-positive HER2 negative breast cancer is being considered as an alternative to failing endocrine therapy [44]. Pemigatinib, infigratinib, fexagratinib, and zoligratinib, for example, are inhibitors for FGFR1/2/3, while lirafugratinib acts against FGFR2 alone [84,85]. A FGFR2 *IIIb*-targeting monoclonal antibody, bemarituzumab (FPA144), is in trials for the treatment of solid tumours [84,86]. However, the FGFR2 isoform switch from the *IIIb* to *IIIc* isoform due to EMT is associated with cancer aggressiveness and advanced clinical stages and may limit the benefits of bemarituzumab [84]. The FGFR-specific inhibitor had a modest benefit in the phase II RADICAL trial for patients with ER-positive breast cancer treated with aromatase inhibitors [87], while the tyrosine kinase inhibitor, Suntinab, was not anti-tumorigenic for patients with FGFR1 or FGFR2 changes in the Targeted Agent and Proliferation Utilization Registry study [88]. The FGFR2 inhibitor, RLY-4008, may be able to selectively target FGFR2 and is being investigated in cholongiocarcinoma [79,89].

## 5. Conclusions

There is interest in FGFR2 as a possible target for therapeutics in various cancers, including breast cancer, increasing the need to understand the effects of FGFR2 on sustaining, and facilitating, the progression of cancer. Mining data from available sources, we determined that *FGFR2* mRNA levels are higher in less aggressive ER-positive tumours and ILC. By contrast, the levels are lowest in HER2-expressing tumours, and lower in carcinomas with IDC. The more aggressive splice isoform (*IIIc*) is found in HER2-positive cancers; whereas the splice isoform (*IIIb*) is found in ER-positive cancers. This suggests that an increased *FGFR2* expression is related to maintaining the epithelial expression of breast cancer, whereas a lower *FGFR2* is expressed in more aggressive forms of breast cancer.

## Figures and Tables

**Figure 1 biology-13-00920-f001:**
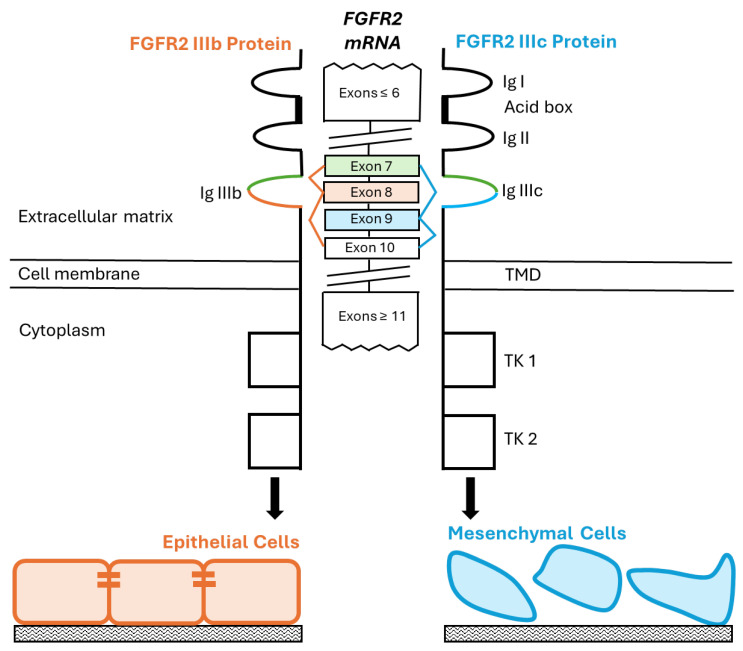
Scheme showing *FGFR2 IIIb* and *IIIc* splice variants. The extracellular domain of the FGFR2 protein contains three Ig-like loops, an acid box between IgI and IgIII, a transmembrane domain (TMD), and intracellular tyrosine kinases (TKs). The Ig-III loop is coded for by exons 7, 8, and 9. *FGFR2* expresses exons 7 and 8 but excludes exon 9 in the Ig-III domain to form *FGFR2 IIIb* in epithelial cells. By contrast, expression of exons 7 and 9 (excluding exon 8) results in *FGFR*2 *IIIc* in the mesenchymal cells. This splice variant is regarded as a hallmark of EMT, with more aggressive carcinomas associated with the *IIIc* splice variant.

**Figure 2 biology-13-00920-f002:**
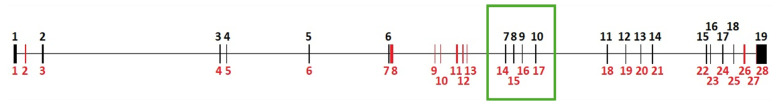
Representative *FGFR2* gene map. The gene map shows all known splicing events derived from TCGA SpliceSeq data correlated to the GRCh38 sequence. Exons are shown in **black**, consistent with the GRCh38 numbering. Alternatively spliced-in regions are shown in **red**, as is the TCGA SpliceSeq numbering. The splice variants of interest were exons 7, 8, 9, and 10 (GRCh38), which correlated to 14, 15, 16, 17.1, and 17.2 (TCGA SpliceSeq), highlighted in the **green** box.

**Figure 3 biology-13-00920-f003:**
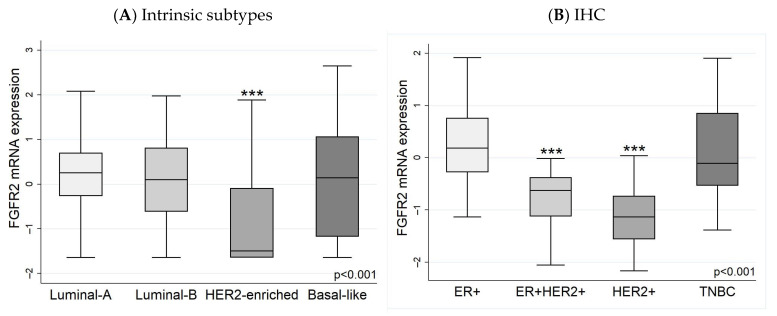
*FGFR2* mRNA expression in (**A**) intrinsic subtypes and (**B**) IHC subtypes. (**A**) *FGFR2* expression was significantly lower in HER2-enriched subtype compared with luminal-A, luminal-B, and basal-like subtypes, which were not significantly different from each other. (**B**) The IHC results are comparable with the intrinsic classifications. The non-HER+ immunohistochemical subtypes (ER+ and TNBC) did not differ from each other but were higher than the HER2-expressing subtypes (*p* < 0.001). The ER+HER2+ and HER2+ subtypes did not differ significantly from each other. *FGFR* mRNA expression was log_2_-transformed. *** *p* < 0.001.

**Figure 4 biology-13-00920-f004:**
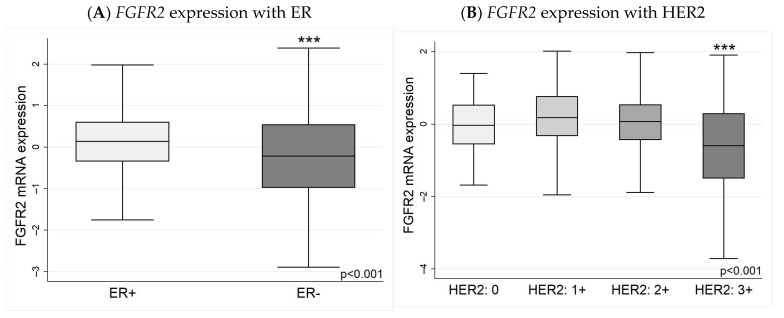
*FGFR2* expression in (**A**) ER-positive and ER-negative tumours and (**B**) HER2 expression. (**A**) *FGFR2* mRNA expression is higher in ER+ than ER− breast cancers. (**B**) HER2 expression is considered negative when there is a value of 0 or 1+, positive at a value of 3+, and equivocal at 2+. Tumours with HER2 of 0, 1+, and 2+ are not significantly different from each other. However, *FGFR2* is significantly downregulated when HER2 has a 3+ value (i.e., HER2-positive) when compared to HER2: 0, 1+, and 2+. *FGFR* mRNA expression was log_2_-transformed. *** *p* < 0.001.

**Figure 5 biology-13-00920-f005:**
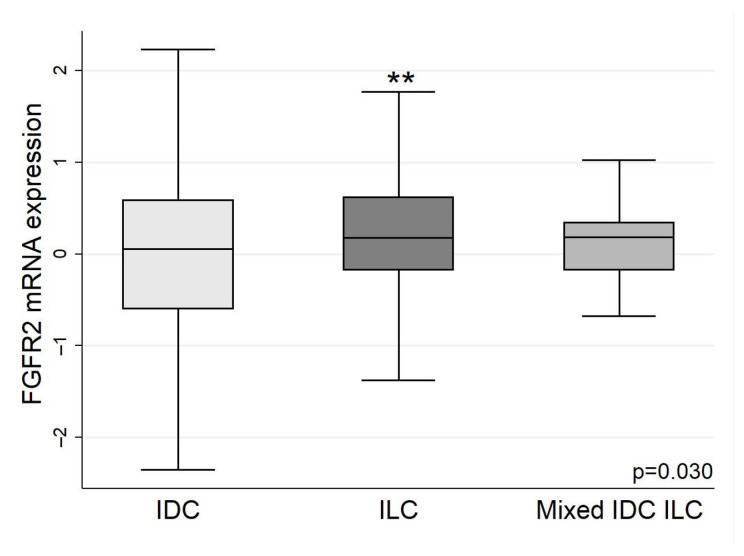
*FGFR2* expression in invasive ductal and invasive lobular carcinomas. *FGFR2* mRNA expression was higher in ILC than IDC tumour types. As expected, people who had both lobular and ductal carcinomas had *FGFR2* expression between ILC and IDC. *FGFR* mRNA in invasive carcinomas presented as log_2_-transformed. ** *p* < 0.01.

**Figure 6 biology-13-00920-f006:**
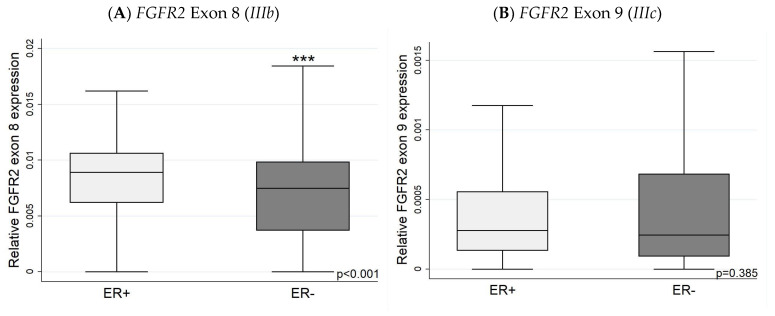
*FGFR2* exon 8 and 9 expression in ER+ and ER− tumours. (**A**) Expression of exon 8 (indictive of the *IIIb* isoform) is greater in ER+ than ER− tumours; while (**B**) exon 9 expression levels do not differ. Exon 8 and exon 9 mRNA expression levels are presented as relative to overall *FGFR2* expression levels. *** *p* < 0.001.

**Figure 7 biology-13-00920-f007:**
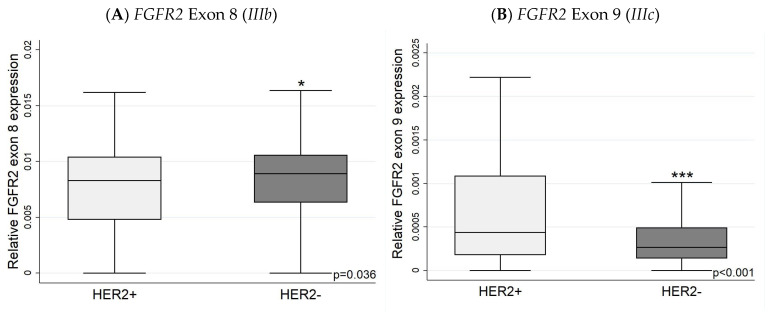
*FGFR2* exon 8 and 9 expression in HER2+ and HER2− tumours. (**A**) Expression of exon 8 (*IIIb*) is higher in HER2− tumours; by contrast, (**B**) exon 9 is significantly higher in HER2+ tumours. Exon 8 and exon 9 mRNA expression levels are relative to overall *FGFR2* expression levels. * *p* < 0.05 *** *p* < 0.001.

**Figure 8 biology-13-00920-f008:**
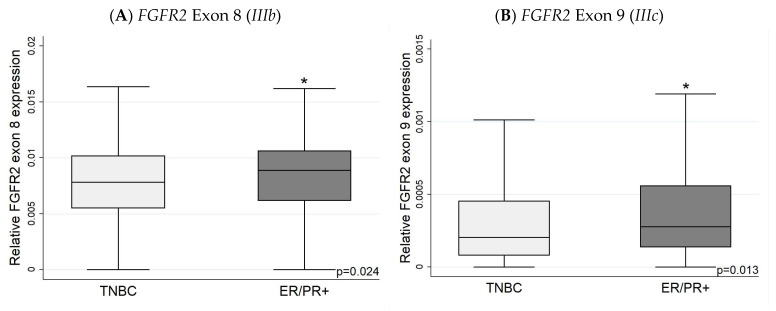
*FGFR2* exon 8 and 9 expression in TNBC and ER/PR+ tumours. The TNBC has lower exon 8 (*IIIb*) (**A**) and lower exon 9 (*IIIc*) (**B**) expression than ER/PR+ tumours. Exon 8 and exon 9 mRNA expression levels presented as relative to overall *FGFR2* expression levels. * *p* < 0.05.

**Figure 9 biology-13-00920-f009:**
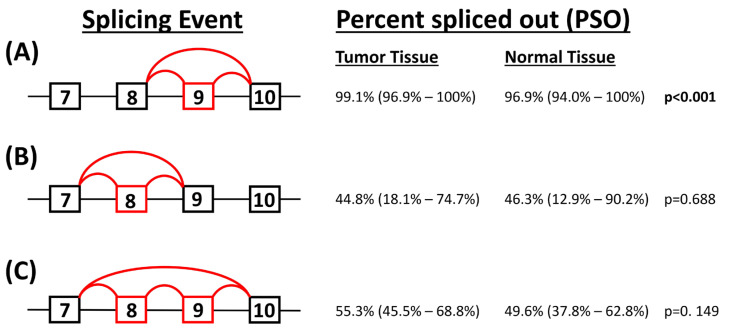
Percent splicing of *FGFR2* exons 8 and 9. Exclusion of exons 8 and 9 was examined using the TCGA SpliceSeq program. Results are presented as the percent spliced out (number of reads not containing the exon of interest divided by the total number of reads covering that region). The skipped exon is shown in red. (**A**) Skipping exon 9, creating a *IIIb* transcript, was covered in 1001 samples from tumour tissue and 97 samples from normal tissue. Exon 9 was excluded in most samples, with significantly more exclusion in tumour tissue compared to normal tissue. (**B**) Skipping exon 8, creating a *IIIc* transcript, was covered in 74 samples from tumour tissue and 29 samples from normal tissue. (**C**) Skipping both exons 8 and 9 was covered in 70 tumour samples and 19 normal tissue samples. All data are presented as medians and interquartile ranges with *p*-values determined using the Mann–Whitney test.

**Table 1 biology-13-00920-t001:** Data transposed from TCGA SpliceSeq for PSI analysis.

as_id	TCGA SpliceSeq Nomenclature	GRCh38 Nomenclature	Schematic Using GRCh38 Nomenclature
as_id 13303	Ex 15–Ex 17.1; Skip Ex 16	Ex 8–Ex 10;Skip Ex 9	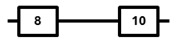
as_id 13305	Ex 14–Ex 16; Skip Ex 15	Ex 7–Ex 9; Skip Ex 8	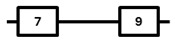
as_id 126925	Ex 14–Ex 17.1; Skip Ex 15 &16	Ex 7–Ex 10; Skip Ex 8 & 9	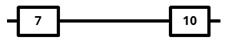

as_id, alternative splicing identification; TCGA, The Cancer Genome Atlas; GRCh38, Genome Reference Consortium Human Build 38; Ex, exon.

## Data Availability

The data were obtained from publicly available databases.

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
