# Peer review of "Lower FGFR2 mRNA Expression and Higher Levels of FGFR2 IIIc in HER2-Positive Breast Cancer"

_biology, 2024, doi:10.3390/biology13110920_

Round 1

Reviewer 1 Report

Comments and Suggestions for Authors

In this manuscript, the authors analyzed published data to examine the association between fibroblast growth factor receptor 2 (FGFR2) total/isoforms expression and breast cancer subtypes. Below are my specific comments/suggestions.

1.      Legends of several figures become in-text paragraphs including Figure 1 (lines 74-80), Figure 2 (lines 153-158), Figure 5 (lines 201-204) and Figure 9 (lines 250-259). Please revise.

2.      Please consider combining the two paragraphs in lines 81-82 and 83-86 to improve the flow of the Introduction section.

3.      There are several font size errors including lines 83-84, 91-93, line 116, line 122, line 124, line 126 and line 142. Please revise.

4.      Line 141 contains syntax error. Please revise.

5.      Please define PAM50 in its first use (line 95).

6.      Please define IDC/ILC in their first use (line 103).

7.      Some Figures have been partially cut out at the top including Figure 6a and Figure 7B. Please revise.

8.      RE: “A monoclonal antibody, bemarituzumab (FPA144) is in trials for treatment of FGFR2 IIIb.” (lines 371-372) This sentence is misleading. Please consider revising (i.e. “A FGFR2 IIIb targeting monoclonal antibody, bemarituzumab (FPA144), is in trials for treatment of solid tumors”).

Comments on the Quality of English Language

There are several syntax errors that require revision.

Author Response

We thank Reviewer one for their comments. The changes that we made regarding reviewer one’s comments have been highlighted in the manuscript.

  1. Legends of several figures become in-text paragraphs including Figure 1 (lines 74-80), Figure 2 (lines 153-158), Figure 5 (lines 201-204) and Figure 9 (lines 250-259). Please revise.

During the format change from the original word document to the style preferred by “Biology”, some of the legends became part of the text. We have checked and corrected all figure legends. We have also aligned the figures to the right so that they are visually in line with the text. In text has been changed for Fig 1: page 3, lines 88-94; Fig 2: page5, lines 185-191; Fig 5: page 7, lines 240-245; Fig 9: pages 9+10, lines 301-310.

  1. Please consider combining the two paragraphs in lines 81-82 and 83-86 to improve the flow of the Introduction section.

We thank the reviewer for the suggestion and have combined the two sentences, as seen on page 4, lines 115-116.

  1. There are several font size errors including lines 83-84, 91-93, line 116, line 122, line 124, line 126 and line 142. Please revise.

Thank you for pointing out the size differences. We have corrected this so that the text is consistently font size 10 and the figure legends are font size 9.These have been corrected on page 4 lines 124-125; 156-157; 161.

  1. Line 141 contains syntax error. Please revise.

We think that this was an error in the original word document, but it has been corrected in the revised manuscript, on page 5, lines 175-176.

  1. Please define PAM50 in its first use (line 95).

Thank you to the reviewer. We missed this acronym and have included it on page 4, line128.

  1. Please define IDC/ILC in their first use (line 103).

Similar to point 5, we missed the defining these acronyms when they were first used. We have updated this on page 4, line 137.

  1. Some Figures have been partially cut out at the top including Figure 6a and Figure 7B. Please revise.

Our apologies for this. We think that the headings moved during the transfer from the first draft, to the style of “Biology”. We have subsequently checked the figure headings for all figures. This has been changed on fig 6: page 8, line 265 and fig 7: page 8, line 273

  1. RE: “A monoclonal antibody, bemarituzumab (FPA144) is in trials for treatment of FGFR2 IIIb.” (lines 371-372) This sentence is misleading. Please consider revising (i.e. “A FGFR2 IIIb targeting monoclonal antibody, bemarituzumab (FPA144), is in trials for treatment of solid tumors”).

Thank you to the reviewer for this comment. We have changed the phrasing to be more accurate, as seen on page 12, lines 420-423.

  1. There are several syntax errors that require revision.

We have gone through the document to keep the presentation as consistent as possible.

Reviewer 2 Report

Comments and Suggestions for Authors

The aim of the analyses conducted by Therese Dix-Peek and colleagues is to determine the level of FGFR2 expression in different molecular subtypes of breast cancer. The authors of the manuscript attempted to explain an important phenomenon in the context of neoplastic diseases and the observed rapid increase in the incidence of breast cancer in the population. After reading the manuscript, I have several comments.

The introduction of the submitted manuscript is unfinished and written too generally. Information on the characteristics of molecular subtypes of breast cancer should be supplemented and the role of FGFR2 in breast cancer should be described more broadly.

When analyzing the level of FGFR2 expression in different molecular subtypes of breast cancer, the authors should take into account different clinical data of patients and distinguish several groups in which they analyze the level of expression of the described gene within a given type of breast cancer, e.g. age, metastases to lymph nodes, average survival, etc., because these data may affect expression and significantly differ between clinical groups within the same molecular subtype of cancer.

Author Response

We thank Reviewer two for their comments and changes made in response to their comments have been highlighted on the revised manuscript.

  1. The introduction of the submitted manuscript is unfinished and written too generally. Information on the characteristics of molecular subtypes of breast cancer should be supplemented and the role of FGFR2 in breast cancer should be described more broadly.

We thank the reviewer for their comments.  We have improved the introduction by adding information on molecular subtypes (pages 1 and 2, lines 37-46), as well as immunohistochemical subtypes (page 2, lines 49-55).

We have added information on the action of FGFR2 expression in breast cancer and the effects of the splice isoforms on the cancer subtypes (page 3, lines 96-111).

  1. When analyzing the level of FGFR2 expression in different molecular subtypes of breast cancer, the authors should take into account different clinical data of patients and distinguish several groups in which they analyze the level of expression of the described gene within a given type of breast cancer, e.g. age, metastases to lymph nodes, average survival, etc., because these data may affect expression and significantly differ between clinical groups within the same molecular subtype of cancer.

We thank the reviewer for their suggestions. We performed a multivariable linear regression with FGFR2 expression levels as our outcome, with independent variables molecular subtype, age, nodal metastases, survival and stage at diagnosis included in the model. Of these, only the HER2-enriched subtypes were found to have a significant association with FGFR2. We also ran a multivariable linear regression on FGFR2 expression with the IHC subtypes, age, and stage of diagnosis as independent variables. In this case, the ER+HER2+ and HER2+ subtypes were found to be significant.

We have included a line in both the materials and methods (page 4, line 145-147) and results (page 6 lines 206-209 and lines 211-213) sections for this analysis, and have presented the analysis in supplementary table S2.

Round 2

Reviewer 2 Report

Comments and Suggestions for Authors

I accept the responses submitted by the authors and recommend the manuscript for publication in "Biology".